# Phase I Dose Escalation Study with Expansion Cohort of the Addition of Nab-Paclitaxel to Capecitabine and Oxaliplatin (CapOx) as First-Line Treatment of Metastatic Esophagogastric Adenocarcinoma (ACTION Study)

**DOI:** 10.3390/cancers11060827

**Published:** 2019-06-14

**Authors:** Sandor Schokker, Stephanie O. van der Woude, Jessy Joy van Kleef, Daan J. van Zoen, Martijn G. H. van Oijen, Banafsche Mearadji, Ludo F. M. Beenen, Charlotte I. Stroes, Cynthia Waasdorp, R. Aarti Jibodh, Aafke Creemers, Sybren L. Meijer, Gerrit K. J. Hooijer, Cornelis J. A. Punt, Maarten F. Bijlsma, Hanneke W. M. van Laarhoven

**Affiliations:** 1Laboratory for Experimental Oncology and Radiobiology, Center for Experimental and Molecular Medicine, Cancer Center Amsterdam, Amsterdam UMC, University of Amsterdam, Meibergdreef 9, 1105AZ Amsterdam, The Netherlands; s.schokker@amc.uva.nl (S.S.); c.i.stroes@amsterdamumc.nl (C.I.S.); c.waasdorp@amc.uva.nl (C.W.); r.a.jibodh@amc.uva.nl (R.A.J.); a.creemers@amsterdamumc.nl (A.C.); m.f.bijlsma@amc.uva.nl (M.F.B.); 2Department of Medical Oncology, Cancer Center Amsterdam, Amsterdam UMC, University of Amsterdam, Meibergdreef 9, 1105AZ Amsterdam, The Netherlands; s.o.vanderwoude@amsterdamumc.nl (S.O.v.d.W.); j.j.vankleef@amsterdamumc.nl (J.J.v.K.); d.j.vanzoen@amc.uva.nl (D.J.v.Z.); m.g.vanoijen@amsterdamumc.nl (M.G.H.v.O.); c.punt@amsterdamumc.nl (C.J.A.P.); 3Department of Radiology, Amsterdam UMC, University of Amsterdam, Meibergdreef 9, 1105AZ Amsterdam, The Netherlands; b.mearadji@amc.uva.nl (B.M.); l.f.beenen@amc.uva.nl (L.F.M.B.); 4Department of Pathology, Amsterdam UMC, University of Amsterdam, Meibergdreef 9, 1105AZ Amsterdam, The Netherlands; s.l.meijer@amc.uva.nl (S.L.M.); g.k.hooijer@amc.uva.nl (G.K.J.H.)

**Keywords:** esophagogastric cancer, phase I clinical trial, nab-paclitaxel, CapOx, stromal cells

## Abstract

First-line triplet chemotherapy including a taxane may prolong survival in patients with metastatic esophagogastric cancer. The added toxicity of the taxane might be minimized by using nab-paclitaxel. The aim of this phase I study was to determine the feasibility of combining nab-paclitaxel with the standard of care in the Netherlands, capecitabine and oxaliplatin (CapOx). Patients with metastatic esophagogastric adenocarcinoma received oxaliplatin 65 mg/m^2^ on days 1 and 8, and capecitabine 1000 mg/m^2^ bid on days 1–14 in a 21-day cycle, with nab-paclitaxel on days 1 and 8 at four dose levels (60, 80, 100, and 120 mg/m^2^, respectively), using a standard 3 + 3 dose escalation phase, followed by a safety expansion cohort. Baseline tissue and serum markers for activated tumor stroma were assessed as biomarkers for response and survival. Twenty-six patients were included. The first two dose-limiting toxicities (i.e., diarrhea and dehydration) occurred at dose level 3. The resulting maximum tolerable dose (MTD) of 80 mg/m^2^ was used in the expansion cohort, but was reduced to 60 mg/m^2^ after three out of eight patients experienced diarrhea grade 3. The objective response rate was 54%. The median progression-free (PFS) and overall survival were 8.0 and 12.8 months, respectively. High baseline serum ADAM12 was associated with a significantly shorter PFS (*p* = 0.011). In conclusion, albeit that the addition of nab-paclitaxel 60 mg/m^2^ to CapOx may be better tolerated than other taxane triplets, relevant toxicity was observed. There is a rationale for preserving taxanes for later-line treatment. ADAM12 is a potential biomarker to predict survival, and warrants further investigation.

## 1. Introduction

The mortality of gastric and esophageal carcinoma remains high, particularly due to the high rate of locally advanced or metastatic disease at diagnosis, the substantial recurrence rate after treatment with curative intent, and the development of treatment resistance [1]. Therefore, there is an urgent need for effective first-line palliative treatment.

First-line treatment usually consists of a doublet with a fluoropyrimidine and a platinum compound, sometimes extended to a triplet, and in the case of a HER-2 positive tumor, trastuzumab can be added [2,3]. The addition of an anthracycline only marginally improves survival, while the addition of a taxane more effectively improves survival, albeit at the cost of increased toxicity [4]. Nab-paclitaxel has a favorable toxicity profile compared to conventional taxanes, and has shown non-inferiority to solvent-based paclitaxel with less toxicity in the second-line treatment of gastric carcinoma [5]. The primary aim of this study was to determine the recommended dose for phase II testing (RP2D) of nab-paclitaxel combined with capecitabine and oxaliplatin (CapOx), and to assess its feasibility in terms of toxicity.

Traditionally, cancer research has focused on the treatment of tumor cells [3,6]. However, there is emerging evidence that the collective of non-tumor cells and material (i.e., tumor stroma) plays a pivotal role in tumor growth, as well as metastasis formation and the development of resistance [7,8]. Thus, the stroma should be taken into consideration when designing novel treatment strategies [9,10,11,12,13,14,15]. Among stromal cells, cancer-associated fibroblasts (CAFs) are of utmost importance, not only because of their relative abundance, but particularly because of their cross-talk with tumor cells [16]. Previous research has shown that CAFs are associated with a worse prognosis in esophageal cancer and precede the deposition of collagen—another main component of the stroma [7,14]. Due to its relative abundance and mechanical properties, the stroma may function as a barrier to cytotoxic treatment [8,9]. It can increase interstitial pressure, with the compression of existing capillaries and the restriction of new vessel formation as a result. In turn, this leads to tumor hypoxia, which is associated with metastatic propensity of tumor cells as well as with treatment resistance, and also limits the effective delivery of drugs to cancer cells [17,18].

Moreover, CAFs secrete cytokines such as IL-6, which may further contribute to treatment resistance [16,19,20]. Our secondary aim was therefore to explore the prognostic importance of activated stroma during the first-line treatment of advanced esophagogastric cancer.

## 2. Results

### 2.1. Patients

Between December 2014 and November 2016, 36 patients were assessed, and 26 eligible patients were enrolled (Figure 1). All patients had metastatic disease, and median age at time of enrolment was 63 years (range 45–75) (Table 1). Of the ten patients that were previously treated with curative intent, the majority had received neoadjuvant chemoradiation with carboplatin, paclitaxel, and fractionated radiation [21], and six had undergone potentially curative surgery (see Appendix A for a detailed description of all included patients with regards to prior treatment).

### 2.2. Safety and Tolerability

#### 2.2.1. Dose Escalation, DLTs, and MTD

At data cut-off on 2 May 2018, 26 patients received at least one cycle of study medication and were all available for safety analysis. Twelve patients were enrolled in the dose-limiting toxicity (DLT) analysis cohort (Figure 2). No DLT was observed at dose levels 1 and 2, but one patient at dose level 2 died outside of the DLT period due to non-neutropenic sepsis and dehydration, suspected to be treatment related. Therefore, dose level 2 was expanded with a further three patients, who did not experience DLT. At dose level 3, two patients experienced DLT; both grade 3 diarrhea, resulting in grade 3 dehydration in one patient. Thus, the MTD was established at 80 mg/m^2^ of nab-paclitaxel and a preplanned 14 patients were to be included in a safety expansion cohort at this dose level. However, after the enrolment of six patients, a further three patients experienced grade 3 diarrhea necessitating hospitalization and dose reductions for both nab-paclitaxel and oxaliplatin. In consultation with the institutional review board, the remaining eight patients of the expansion cohort were treated at dose level 1. At this dose level, grade 3 diarrhea occurred in one patient (9%) and the RP2D was there for eestablished at 60 mg/m^2^.

#### 2.2.2. Treatment-Related Adverse Events

Because of the substantial difference in toxicity between dose levels (particularly grade 3 diarrhea and dehydration), treatment-related adverse events are presented for dose levels 1,2, and 3 separately (Table 2).

A total of 18 serious adverse events occurred in 12 (46%) patients, 10 of which were suspected to be treatment related: 4 in the 60 mg/m^2^ cohort and 14 in the 80 and 100 mg/m^2^ cohorts. Eight patients experienced diarrhea and/or vomiting, in five cases complicated by dehydration. One patient was admitted for an evaluation of chest pain after taking capecitabine, and one patient died of non-neutropenic sepsis. Treatment-related adverse events resulting in discontinuation of study medication were bone marrow toxicity and non-neutropenic sepsis (*n* = 1). One patient had a treatment-unrelated chylothorax for which study medication was discontinued.

### 2.3. Drug Exposure and Clinical Activity

Twenty patients (77%) completed six cycles of combination therapy, and five patients (19%) were eligible for reintroduction of triple treatment after progression on capecitabine monotherapy (Table 3, Figure 3). Median dose intensity during the initial six cycles was 7424 mg/m^2^/week (89.2%) for capecitabine, 40.2 mg/m^2^/week (84.6%) for oxaliplatin, and 43.5 mg/m^2^/week (83.9%) for nab-paclitaxel.

The objective response rate was 54%, with best responses being a complete response in one, partial response in 13 (50%), stable disease in nine (34%), and progressive disease in two patients (8%) (Figure 3). Radiological responses in dose level 1 appeared similar to those in dose levels 2 and 3 (Figure 3). One patient was non-evaluable because of clinical deterioration before a response evaluation CT was performed. Median duration of response was 7.8 months. Half of the patients with progressive disease (12/24) received a subsequent line of palliative treatment (Appendix A). Survival data were calculated for exploratory purposes only. Median progression-free survival (PFS) and overall survival (OS) were 8.0 months and 12.8 months, respectively, with a median follow-up of 13.3 months (range 1.2 to 38.7).

### 2.4. Patient-Reported Health-Related Quality of Life and Neurotoxicity

Twenty-four patients completed the baseline questionnaires before start of treatment (Appendix A and Table 4). Compliance of follow-up questionnaires was high, and all patients but one was on active CapOx–nab-paclitaxel treatment up until cycle 7 (Appendix A). Thereafter, all patients were on capecitabine monotherapy. Four patients that were eligible for reintroduction completed questionnaires before reintroduction and after the first cycle of reintroduction. Mean global health, functioning scores, and symptom scores of the QLQ-C30 questionnaire remained relatively stable during treatment as well as after cessation of triple therapy (Figure 4). The other functioning and symptom scores also remained relatively stable (Appendix A). However, self-reported sensory neuropathy showed an increase from a mean score of 1.39 at baseline to 28.8 after 9 cycles, subsequently decreasing to 17.0 after 15 cycles. The motor and autonomic neuropathy scores demonstrated a similar pattern, albeit with a smaller amplitude. Median time to deterioration of health-related quality of life (HRQoL) was 8.0 months, with only three patients having a definitive 10-point drop in global health score. The remainder had progression or death as their first event.

### 2.5. Biomarkers

Histological biopsies were available for 14 patients, of whom baseline characteristics were similar to the entire cohort (Appendix A). Activated stroma (measured by alpha smooth muscle actin (αSMA) staining) and its product collagen (measured by picrosirius red staining) were not associated with PFS or OS (*p* = 0.63 and 0.98 for αSMA, and *p* = 0.51 and 0.75 for picrosirius red staining, respectively; see Appendix A for representative immunohistochemistry pictures). Baseline serum to assess serum activated stroma markers IL-6 and ADAM12 was available in all patients. Median baseline IL-6 was 0 µL/mL (range 0–61.94), with 17 patients having undetectable IL-6 levels. These patients tended to have a longer median OS time (*p* = 0.10), and patients who had undetectable IL-6 levels throughout treatment had a longer median PFS time (*p* = 0.06, HR 0.44 (CI (confidence interval) 0.18–1.06), Appendix A). Median baseline ADAM12 concentration was 242.5 pg/mL and a high baseline ADAM12 level (dichotomized using Cutoff Finder at 190 pg/mL) was associated with a significantly shorter median PFS time (*p* = 0.008, HR 3.06 (CI 1.29–7.26) Figure 5) but not with median OS time (*p* = 0.096). Baseline ADAM12 levels were also significantly higher in patients with progressive or non-evaluable disease as their best responses compared to partial responders (*p* = 0.002, Appendix A). The median ADAM12 values between the pre-treated and previously untreated groups were statistically similar (pre-treated 171 pg/mL and previously untreated 348 pg/mL, Mann–Whitney *p* = 0.90). Median ADAM12 values between patients who received prior surgery were also statistically similar to those of patients who had not received prior surgery (187 pg/mL vs. 243 pg/mL, respectively, Mann–Whitney *p* = 0.88).

Baseline tumor burden (median 111.16 mL; range 2.34–3475.45 mL) was not correlated with baseline ADAM12 and IL-6 concentrations (*p* = 0.29 and *p* = 0.28, respectively), indicating that the poor survival of patients with high baseline ADAM12 levels was not merely due to a higher tumor burden (see Appendix A for representative images for primary tumor and metastases segmentation).

## 3. Discussion

In order to find a taxane-containing triplet with acceptable toxicity for the palliative treatment of metastatic esophagogastric cancer patients, we combined capecitabine and oxaliplatin with nab-paclitaxel. We established the maximum tolerated dose (MTD) of nab-paclitaxel at 80 mg/m^2^. However, with continuing treatment beyond the DLT period, we observed excessive toxicity. Toxicity was more manageable at 60 mg/m^2^, and this dose was established as RP2D. Median PFS and OS times were promising with 8.0 and 12.3 months, respectively. Compared with other taxane triplets in similar Western populations, there was less hematologic toxicity at the RP2D, especially grade 3/4 neutropenia, while gastro-intestinal toxicity appeared similar [22,23,24,25,26,27,28,29,30]. Additionally, higher-grade polyneuropathy was not observed (0%). However, compared with CapOx alone, substantially more diarrhea, fatigue, anorexia, and peripheral neuropathy were observed [31].

Baseline global health score was high with a mean score of 70.5, compared to other first-line studies, possibly reflecting the good clinical condition of our patients at the start of treatment [23,25,28]. The time to deterioration of global health score of 8.0 months is remarkably longer compared to previously reported taxane triplets and fluoropyrimidine doublets [32,33,34]. It may be hypothesized that the good baseline clinical condition of our patients contributed to the relatively good PFS and OS that was observed in this study. Indeed, our PFS and OS results were favorable compared to those of fluoropyrimidine doublets, although the absolute improvement in survival was modest [31,35,36,37,38,39]. The question remains as to whether physicians and patients consider the toxicity of a taxane triplet in the first line of treatment acceptable, even in a modified schedule or albumin-bound, when OS is not substantially improved compared to doublet therapy. This is especially important in an era where second-line taxane-containing treatment is becoming standard of care [40]. Although randomized data are lacking, it may be hypothesized that the sequential therapy of a fluoropyrimidine doublet followed by taxane-containing second-line treatment is as effective as a first-line containing triplet, while overall toxicity is less [41]. In fact, in a recent Japanese phase III trial a numerically longer OS was observed in the fluoropyrimidine doublet arm, compared to the taxane containing triplet arm (15.3 vs. 14.2 months), possibly pertaining to the possibility of giving a taxane in later lines [42]. Unfortunately, the important question of the optimal sequence of chemotherapy will remain unanswered in the near future [43,44].

In contrast to previous work, in our study activated stroma as measured by αSMA staining was not associated with PFS or OS [7]. This could be due to the limited availability of tumor tissue. Serum was available for all patients, and especially ADAM12 demonstrated potential prognostic value as non-inflammation-related serum-borne marker for activated stroma. A similar prognostic impact of ADAM12 has been observed in patients with resectable or metastatic pancreatic cancer [45]. We demonstrated ADAM12 and IL-6 levels to be unrelated to tumor burden, solidifying the value of these markers in assessing the activation status of stroma, as opposed to merely reflecting total tumor burden. This prognostic, and perhaps even predictive, value of ADAM12 needs to be further studied in larger cohorts with the possibility of multivariate testing—the lack of which is a clear limitation of our present study and underscores the exploratory nature of our findings. Furthermore, the heterogeneity of the patients studied in our trial is a limitation. This pertains not only to the primary site (esophagus, gastro-esophageal junction, and stomach), but also to previous anti-cancer treatment including prior surgery. Relatively more patients had a prior esophagectomy compared to a gastrectomy, which may interfere with survival and susceptibility to toxicity, but to the best of our knowledge there are no definitive data on this difference. The small number of patients in a phase I trial hampers further analyses of specific subgroups. However, in clinical practice, both pretreated and untreated patients and both esophageal and gastric cancer patients tend to receive the same first-line palliative treatment in the Netherlands based on the results of the REAL-2 study, where these groups were studied collectively [3]. One can argue the biomarker levels of pretreated patients could be lower than previously untreated patients because of lack of the primary tumor and its stroma; this has to be studied in larger cohorts.

## 4. Materials and Methods

### 4.1. Study Population

Patients aged ≥18 years with pathologically confirmed metastatic or irresectable carcinoma of the stomach or esophagus, Eastern Cooperative Oncology Group (ECOG, Philadelphia, PA, USA) performance status 0 or 1, measurable or evaluable disease according to RECIST 1.1, and adequate bone marrow and organ function were eligible for inclusion (defined by the following laboratory values: absolute neutrophil count (ANC) > 1.5 × 10^9^/L, hemoglobin (Hgb) > 9.0 g/dL (5.6 mmol/L), platelets > 100 × 10^9^/L, serum total bilirubin within ≤ 1.5 × ULN (upper limit of normal); or total bilirubin < 3.0 × ULN with direct bilirubin within normal range in patients with well-documented Gilbert’s syndrome, international normalized ratio (INR) < 1.5 (unless the patient used vitamin K antagonists), serum creatinine < 1.5 × ULN or creatinine clearance > 50 mL/min/1.73 m^2^, alanine aminotransferase (AST) and aspartate aminotransferase (ALT) within normal range (or <3.0 × ULN if liver metastases were present). HER2+ patients were eligible when treatment with trastuzumab was contraindicated.

The study was done in accordance with the ethics principles of the Declaration of Helsinki and the International Council on Harmonization Guidelines on Good Clinical Practice. The study protocol was approved by the Amsterdam UMC Institutional Review Board (project identification code 2014_272). Written informed consent was obtained from all patients. The trial was registered with ClinicalTrials.gov (NCT02273713) and EudraCT (2014-001333-88).

### 4.2. Study Design

This was an open-label, single-center, investigator-initiated phase I dose escalation study with expansion cohort. All patients were enrolled and treated at the Amsterdam UMC, location Amsterdam Medical Center, starting in November 2014, and follow-up remains ongoing. Patients were enrolled in a standard 3 + 3 dose escalation phase followed by a safety expansion cohort to assess four dose levels of nab-paclitaxel (dose level 1: 60 mg/m^2^; dose level 2: 80 mg/m^2^, dose level 3: 100 mg/m^2^, dose level 4: 120 mg/m^2^ on days 1 and 8) combined with capecitabine (1000 mg/m^2^ twice daily on days 1–14) and oxaliplatin (65 mg/m^2^ on days 1 and 8) in a 21-day cycle. Since nab-paclitaxel may achieve higher efficacy and a better toxicity profile using a weekly administration [5,46], we administered nab-paclitaxel on days 1 and 8 of the 21-day cycle (see Appendix A for the dose escalation scheme). To allow for optimal interaction, oxaliplatin was administered on the same days as nab-paclitaxel [47]. The resulting regimen had the same dose intensities of capecitabine and oxaliplatin as previously described in the REAL-2 study [3]. The minimum treatment and safety evaluation requirements for dose escalation were met if the patient had been treated with at least 75% of the assigned treatment of 21 days following the first dose and completed the safety evaluations (see Appendix A for escalation decision rules). Patients were treated with a maximum of six cycles of combination therapy, after which patients without progression continued with capecitabine monotherapy, analogous to conventional first-line CapOx treatment. Patients with progression of disease after more than 3 months of capecitabine monotherapy were eligible for reintroduction of oxaliplatin and nab-paclitaxel.

### 4.3. Study Assessments

The phase I primary endpoint was dose-limiting toxicity and MTD, with the latter being further studied in a safety expansion cohort to have 20 patients treated at MTD. Secondary endpoints were treatment-related adverse events, response rate according to the response evaluation criteria in solid tumors (RECIST) version 1.1 with evaluation CT scans every three cycles, progression-free survival (PFS), overall survival (OS), and self-reported neurotoxicity and health-related quality of life. Exploratory endpoints were the relation between response to treatment and relative abundance of stroma in metastatic tumor tissue, as well as serum activated stroma markers (ADAM12 and IL-6). We followed the reporting recommendations for tumor marker prognostic studies (REMARK) where possible [48].

#### 4.3.1. Toxicity

Toxicity was measured and graded using the National Cancer Institute Common Terminology Criteria for Adverse Events version 4.03 (CTCAE). Any DLT was considered to be related to nab-paclitaxel alone or in combination with oxaliplatin and capecitabine, and unrelated to disease progression, inter-current illness, or concomitant medications. DLT was evaluated after the first cycle and was defined as: grade 3 or 4 non-hematological toxicity; febrile neutropenia > 5 days; grade 4 neutropenia more than 7 days (absolute neutrophil count < 0.5 × 10^9^/L); thrombocytopenia (platelet count < 25 × 10^9^/L); grade ≥ 2 infection or delay of treatment for more than two weeks due to side effects.

#### 4.3.2. Self-Reported Health-Related Quality of Life and Neurotoxicity

Health-related quality of life was assessed with the European Organisation for Research and Treatment of Cancer (EORTC) Quality of Life Questionnaire (QLQ)-C30, and neurotoxicity with the self-reported EORTC QLQ-Chemotherapy Induced Peripheral Neuropathy (CIPN)20 questionnaire. Items were scaled and scored according to the recommended EORTC procedures [49,50]. Domains and single items ranged from 0 to 100 with high function scores denoting a high level of functioning and high symptom scores denoting a high level of symptoms. Questionnaires were sent by mail, and returned before commencement of chemotherapy, before start of the second and fourth cycle, and subsequently with a three-cycle interval.

#### 4.3.3. Biomarkers

Four-micron slides of biopsies from metastatic sites were stained with Hematoxylin & Eosin (H&E), alpha smooth muscle actin (αSMA, DAKO M0851, 1:800 dilution) for the staining of activated fibroblasts, and picrosirius red for collagen staining. Tumor and stroma were identified by an experienced pathologist blinded for outcome (SLM). Picrosirius red and αSMA staining were quantified as percentage of total tumor and stroma using ImageJ software (picrosirius red: RGB, red channel, threshold 230; αSMA: H DAB, brown channel, threshold 190). Serum samples were obtained prior to initial dosing by centrifugation of blood for 10 min at 1300 RPM, and stored at −80 °C until analysis. ADAM12 and IL-6 levels were determined in mono using the human ADAM12 and IL-6 ELISA kits (DuoSet, R&D Systems, Minneapolis, MN, USA), according to manufacturer’s recommendations.

#### 4.3.4. Tumor Volume Assessment

In order to investigate the independent prognostic value of ADAM12 and IL-6, a post-hoc correlation with total tumor burden measured on baseline CT scans was determined. Baseline CT scans were used to establish baseline tumor volume. Assessment of lesions was done by SS and CIS, under the supervision of an experienced radiologist blinded for outcome (BM, 12 years of experience). Semi-automated software (MM oncology, Syngo Via, Siemens Healthineers, Forchheim, Germany) was used for volume calculations (in milliliters) under the supervision of another experienced radiologist blinded for outcome (LFMB, 14 years of experiencewith manual correction if necessary. Primary lesions were included when measurable or evaluable, as well as pathologically enlarged lymph nodes and distant metastases, all according to RECIST 1.1.

### 4.4. Statistical Analysis

Adverse events were reported in frequency tables and analyzed for all patients who received at least one dose of nab-paclitaxel, capecitabine, and oxaliplatin. All time-to-event definitions started with the first dose of study medication. Progression was defined as radiological progression per RECIST 1.1 or all-cause mortality. The Kaplan–Meier method was used to determine median PFS and OS as well as time to deterioration of HRQoL, defined as a definite drop of 10 points on the global health score of the EORTC QLQ-C30 questionnaire, progression, or death [51]. For Cox proportional hazard analyses, variables were dichotomized by the mean when a variable was normally distributed, by median in the case of a non-normal distribution, or with a cut-off value derived from the biomarker-focused web application Cutoff Finder when more than 20 cases were available, as previously published [52]. Statistical analyses were performed with SPSS version 24 (IBM, Armonk, NY, USA).

## 5. Conclusions

We established the RP2D of nab-paclitaxel combined with CapOx at 60 mg/m^2^ on days 1 and 8 of a 21-day cycle. Although this combination may be better tolerated than other taxane triplets, relevant toxicity was observed. Taxanes may better be preserved for later-line treatment. The serum stromal marker ADAM12 is a potential biomarker to predict survival, and warrants further investigation.

## Figures and Tables

**Figure 1 cancers-11-00827-f001:**
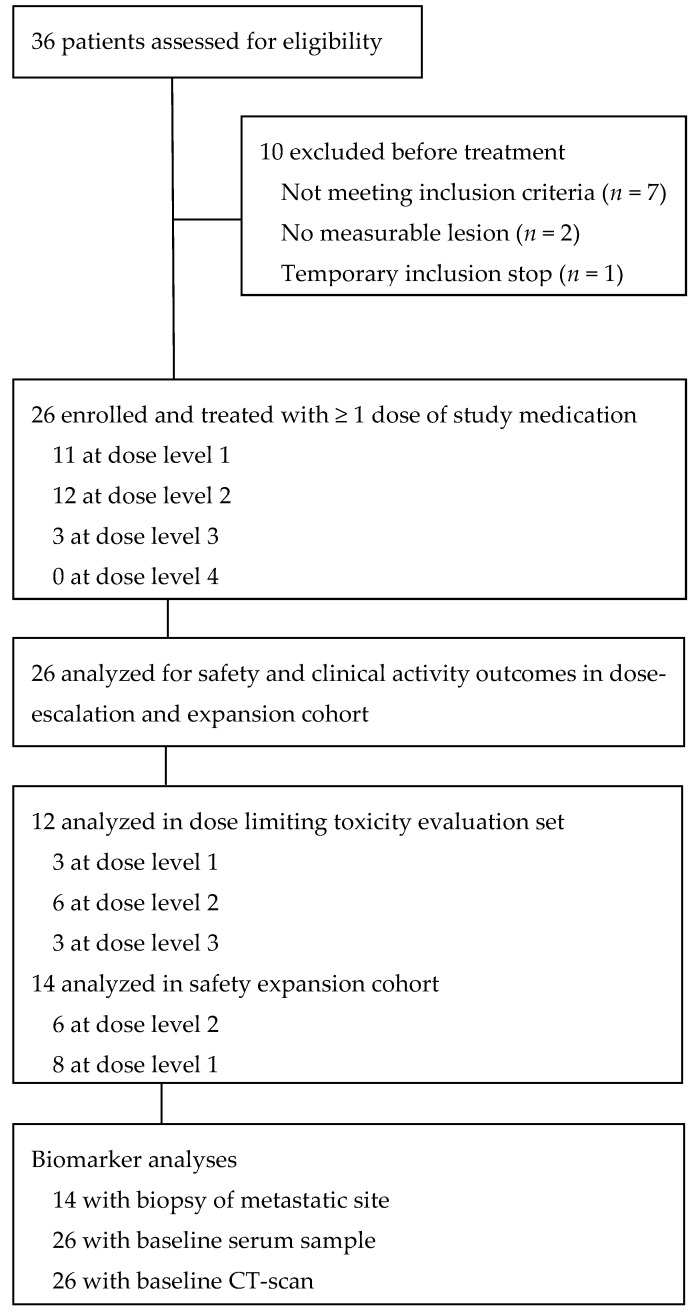
CONSORT flow diagram.

**Figure 2 cancers-11-00827-f002:**
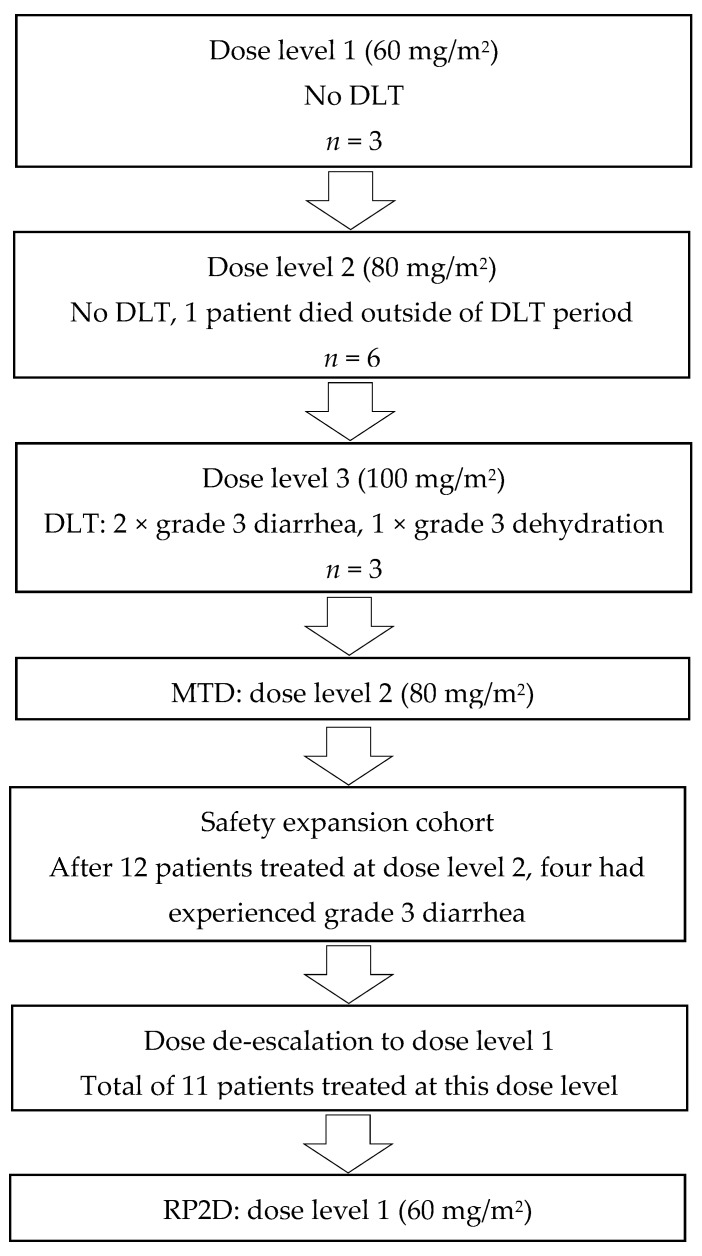
Dose escalation and dose limiting toxicity. Flow diagram of the number of patients per dose level, and the choice for subsequent dose levels based on the occurrence of DLTs and toxicity. DLT: dose-limiting toxicity; MTD: maximum tolerated dose; RP2D: recommended phase II dose.

**Figure 3 cancers-11-00827-f003:**
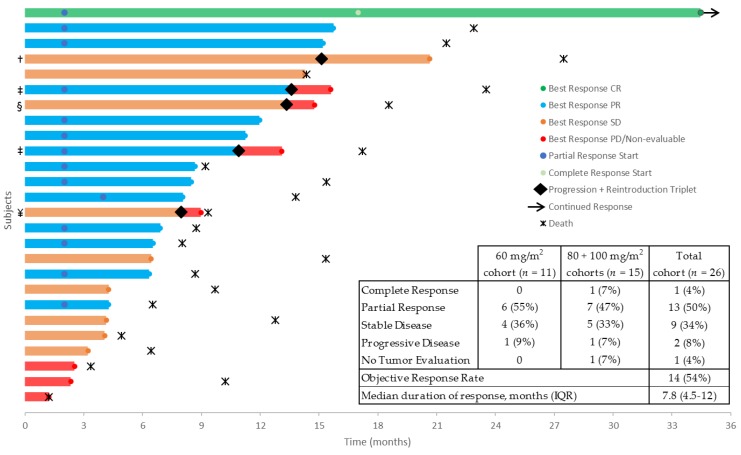
Swimmer plot of radiological response and duration of response. CR: complete response; PR: partial response; SD: stable disease; PD: progressive disease. Patients eligible for reintroduction: † completed six more cycles of capecitabine and oxaliplatin (CapOx)–nab-paclitaxel, progressive disease after three cycles of capecitabine monotherapy; ‡ progressive disease after three cycles; § progressive disease after two cycles; ¥ admitted with pain and malaise one day after reintroduction, remaining off therapy until official diagnosis of progressive disease shortly thereafter.

**Figure 4 cancers-11-00827-f004:**
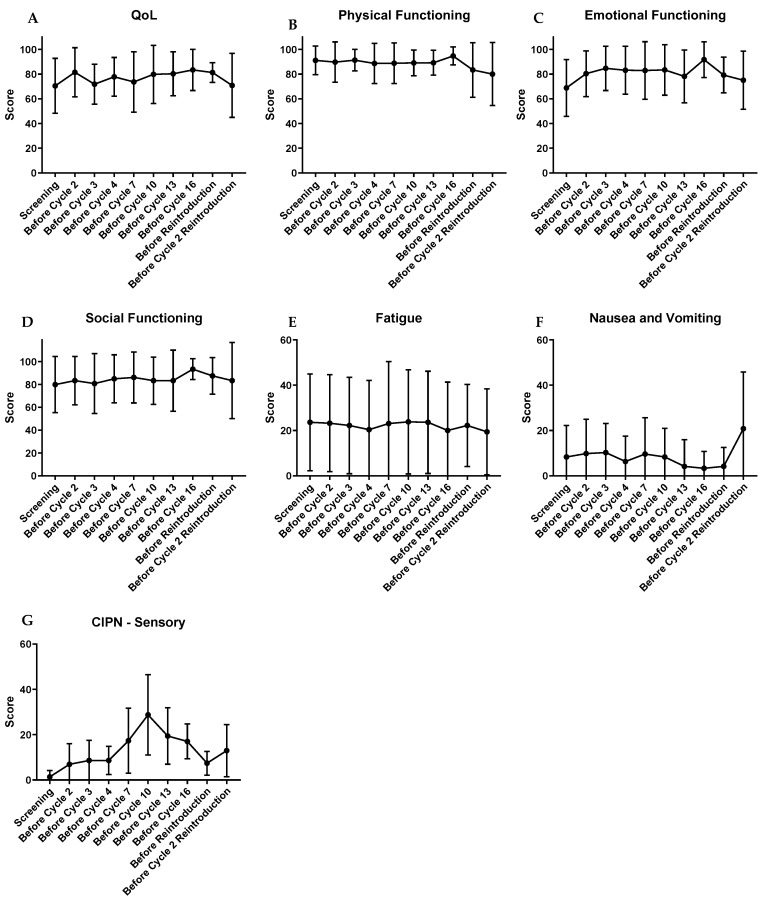
Health related quality of life and neurotoxicity. Domains and single items range from 0 to 100 with high function scores denoting high level of functioning and high symptom scores denoting a high level of symptoms. CIPN: Chemotherapy Induced Peripheral Neuropathy.

**Figure 5 cancers-11-00827-f005:**
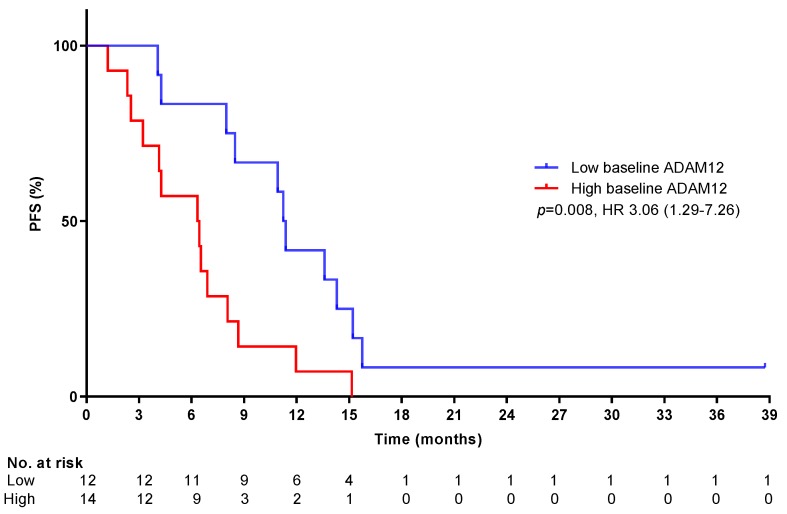
Progression-free survival (PFS) of patients with high versus low baseline ADAM12. Kaplan–Meier curve of patients with high baseline ADAM12 concentration vs. low baseline ADAM12 concentrations, dichotomized at 190 pg/mL (Cutoff Finder). *p* = 0.008 (log-rank test), HR 3.06 (CI 1.29–7.26) (Cox-regression).

**Table 1 cancers-11-00827-t001:** Baseline characteristics.

Baseline Characteristics	*n* (%)/Median (Range)
Median Age at Start Study	63 (45–75)
Sex	
Male	23 (89%)
Female	3 (11%)
ECOG performance status	
0	20 (77%)
1	6 (23%)
Primary tumor location	
Esophagus	9 (35%)
Gastro-esophageal junction	13 (50%)
Stomach	4 (15%)
Disease status	
Synchronous metastases	16 (62%)
Metachronous metastases	10 (39%)
Number of organs involved	
1	10 (39%)
2	10 (39%)
3	3 (11%)
4	3 (11%)
Organs involved	
Lymph nodes	18 (69%)
Liver	12 (46%)
Lung	6 (23%)
Bone	3 (12%)
Peritoneum	3 (12%)
Locoregional recurrence	2 (7%)
Other *	5 (19%)
Prior (neo)adjuvant treatment	
Previous radiotherapy	9 (35%)
Previous chemotherapy **	10 (39%)
Previous gastrectomy	1 (4%)
Previous esophagectomy	5 (19%)
Median time since last (neo-)adjuvant chemotherapy in months (range)	9.8 (1.5–31.7)

* other metastatic sites being adrenals (*n* = 2), testis (*n* = 1), spleen (*n* = 1), rectus abdominus (*n* = 1). ** neoadjuvant chemoradiotherapy with carboplatin and paclitaxel for esophageal carcinoma patients, and perioperative epirubicin, cisplatin, and capecitabine for a gastric cancer patient. ECOG: Eastern Cooperative Oncology Group.

**Table 2 cancers-11-00827-t002:** Treatment-related adverse events.

Adverse Event	60 mg/m^2^ (*n* = 11)		80–100 mg/m^2^ (*n* = 15)	
	Grade 1/2	Grade 3/4/5	Grade 1/2	Grade 3/4/5
**Hematological**	***n* (%)**	***n* (%)**	***n* (%)**	***n* (%)**
Thrombocytopenia	0	1 (9%)	1 (6%)	1 (6%)
Leukocytopenia	0	0	0	1 (6%)
Non-febrile neutropenia	0	2 (18%)	1 (6%)	3 (20%)
**Non-Hematological**	***n* (%)**	***n* (%)**	***n* (%)**	***n* (%)**
Peripheral sensory neuropathy	9 (82%)	0	15 (100%)	0
Fatigue	8 (73%)	0	12 (80%)	0
Nausea	7 (64%)	1 (9)	10 (66%)	1 (6%)
Diarrhea	8 (73%)	1 (9)	9 (60%)	5 (33%)
Dysgeusia	6 (55%)	0	9 (60%)	0
Alopecia	3 (27%)	0	8 (53%)	0
Vomiting	4 (36%)	1 (9)	8 (53%)	2 (13%)
Anorexia	6 (55%)	1 (9)	7 (47%)	0
Oral mucositis	1 (9%)	0	4 (26%)	1 (6%)
Pain	3 (27%)	0	4 (26%)	0
Dehydration	0	0	0	4 (26%)
Fever	0	0	3 (20%)	0
Constipation	3 (27%)	0	2 (13%)	0
Cough	0	0	2 (13%)	0
Dysphagia	0	0	2 (13%)	0
Flu-like symptoms	0	0	2 (13%)	0
Malaise	2 (18%)	0	2 (13%)	0
Non-cardiac chest pain	0	0	2 (13%)	0
Palmar-plantar erythrodysesthesia syndrome	1 (9%)	0	2 (13%)	0
Maculopapular rash	0	0	2 (13%)	0
Hypertension	0	0	0	1 (6%)
Sepsis	0	0	0	1 (6%)
Abdominal pain	1 (9%)	0	1 (6%)	0
Dizziness	1 (9%)	0	1 (6%)	0
Thromboembolic event	1 (9%)	0	1 (6%)	0

Treatment-related adverse events occurring at any grade in >1 patient or at grade ≥ 3 in any patient. Only the highest grade of an adverse event per patient is reported.

**Table 3 cancers-11-00827-t003:** Drug exposure.

Drug Exposure	*n* (%)/Median (Range)
Median number of cycles ACTION (range)	6 (2–6)
Patients completing 6 cycles	20 (77%)
Reasons for premature study termination *	
Toxicity	4 (15%)
Progressive disease	2 (8%)
Patients continuing capecitabine monotherapy	18 (69%)
Median number of cycles capecitabine monotherapy (range)	7 (2–43)
Patients eligible for reintroduction	5 (19%)
Median dose intensity oxaliplatin (range) **	40.2 (22.7–67.4)
Dose intensity oxaliplatin in %	84.6%
Median dose intensity capecitabine (range) **	7424 (4530–9672)
Dose intensity capecitabine in %	89.2%
Median dose intensity nab-paclitaxel (range) **	43.5 (25.7–82.9)
Dose intensity nab-paclitaxel in %, total cohort	83.9%
60 mg/m^2^ cohort	37.5 (25.7–46.0)
Dose intensity in %	87.8%
80 mg/m^2^ cohort	47.6 (27.7–82.9)
Dose intensity in %	88.3%
100 mg/m^2^ cohort	47.3 (41.1–64.6)
Dose intensity in %	78.9%

* Four patients stopped treatment in the course of the six cycles due to toxicities, of whom two died shortly thereafter; two patients were progressive after three and five cycles, respectively. ** Dose intensity in mg/m^2^/week was calculated for the initial cycles of ACTION therapy and consists of the cumulative dose divided by the number of weeks the patient was treated with study medication, up to a maximum of six cycles. Dose intensity in % was calculated as percentage of the theoretical maximum dose intensity. Nab-paclitaxel dose intensity was calculated for the total cohort and the respective dose level cohorts separately.

**Table 4 cancers-11-00827-t004:** Baseline health-related quality of life (HRQoL) and neurotoxicity.

Baseline HRQoL (*n* = 24)	Mean (SD)	Median (IQR)
Global QoL/Global Health Score †‡	70.5 (22.2)	75 (66.6–83.3)
Functioning scales †‡		
Physical functioning	91.1 (11.6)	93.3 (86.7–100)
Role functioning	81.9 (23)	91.7 (66.7–100)
Emotional functioning	68.8 (23)	66.7 (52.1–89.6)
Cognitive functioning	93.1 (12.9)	100 (83.3–100)
Social functioning	79.9 (24.6)	91.7 (66.7–100)
Symptom scales †		
Fatigue	23.6 (21.3)	22.2 (2.8–33.3)
Nausea and vomiting	8.3 (13.9)	0 (0–16.7)
Pain	18.8 (25.2)	8.3 (0–33.3)
Dyspnea	13.9 (19.5)	0 (0–33.3)
Insomnia	25 (31.5)	0 (0–58.3)
Appetite loss	22.2 (27.2)	0 (0–33.3)
Constipation	16.7 (22)	0 (0–33.3)
Diarrhea	22.2 (28.9)	0 (0–33.3)
Financial difficulties	6.9 (17)	0 (0–0)
Neuropathy scales §		
Sensory Neuropathy	1.4 (2.9)	0 (0–2.8)
Motor Neuropathy	1 (2.4)	0 (0–0)
Autonomic Neuropathy	4.2 (8.9)	0 (0–0)

EORTC: European Organisation for Research and Treatment of Cancer. † EORTC QLQ-C30. ‡ Global quality of life or functional scales (scale range 0–100): high score = high level of functioning. § EORTC QLQ-CIPN20. Symptom scales or single items (scale range 0–100): high score = high level of symptoms or problems.

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
