# Peer review of "Phase I Dose Escalation Study with Expansion Cohort of the Addition of Nab-Paclitaxel to Capecitabine and Oxaliplatin (CapOx) as First-Line Treatment of Metastatic Esophagogastric Adenocarcinoma (ACTION Study)"

_cancers, 2019, doi:10.3390/cancers11060827_

Round 1
Reviewer 1 Report
Major Comments
1. In the “Introduction Section” it is not clear the background that support the study of the prognostic importance of activated stroma during first-line chemotherapy of advanced esophagogastric cancer. In my opinion in a phase 1 study there is no rationale to study a marker of prognosis, furthermore the analyzed series is too small for evaluate a biomarker. I suggest to delete all the biomarkers section.
2. Biopsies of metastatic sites are not described. Are they US or CT needle guided? Each biopsied metastatic site should be detailed in a table.
3. Data of OS and PFS should be deleted in that they are typical end points of phase 2 and 3 studies, again the analyzed series of patients is too small for this end points.
4. In the “Materials and Methods Section” immunohistochemistry for the detection of “alpha smooth muscle actin (αSMA, DAKO M0851, 1:800 dilution) for staining of activated fibroblasts” is not detailed, there is no negative control in the experiments. In “Figure S2” magnification is lacking, the micrometric line bar is lacking. These are important bias of the experiments.
Minor Comments
1. In the “Introduction Section” the Authors should specify that first line treatment usually consist of a doublet with a fluoropyrimidine and a platinum compound and with special regard to gastric cancer if the tumour is HER-2 positive trastuzumab should be added.
2. At line 73 should be specified how many patients had received neoadjuvant chemoradiation with carboplatin, paclitaxel and fractionated radiation.
3. At line 74 the sentence “and six had undergone curative surgery” should be changed in “and six had undergone to potentially curative surgery”.
Author Response
Response to Reviewer 1 Comments
Point 1: In the “Introduction Section” it is not clear the background that support the study of the prognostic importance of activated stroma during first-line chemotherapy of advanced esophagogastric cancer. In my opinion in a phase 1 study there is no rationale to study a marker of prognosis, furthermore the analyzed series is too small for evaluate a biomarker. I suggest to delete all the biomarkers section.
Response 1: We thank the esteemed reviewer for this remark and apologize that the potential prognostic impact of activated stroma was not sufficiently clear. We have therefore added the following text in the introduction:
“Traditionally, cancer research has focused on treatment of tumor cells. However, there is emerging evidence that the collective of non-tumor cells and material – i.e. tumor stroma - plays a pivotal role in tumor growth, as well as metastasis formation and development of resistance. Thus, when designing novel treatment strategies, the stroma should be taken into consideration. Among stromal cells, cancer-associated fibroblasts (CAFs) are of utmost importance, not only because of their relative abundance, but particularly because of their cross talk with tumor cells. Previous research has shown that CAFs are associated with a worse prognosis in esophageal cancer and precede the deposition of collagen, another main component of the stroma. Due to its relative abundance and mechanical properties, the stroma may function as a barrier to cytotoxic treatment. It can increase interstitial pressure, with compression of existing capillaries and restriction of new vessel formation as a result. This, in turn, leads to tumour hypoxia, which is associated with metastatic propensity of tumor cells as well as with treatment resistance, and also limits the effective delivery of drugs to cancer cells.”
We agree that this study is not adequately powered to demonstrate true prognostic value of a biomarker. Nevertheless, we added these exploratory analyses because we believe it is important to investigate potential biomarkers in early phases of clinical trials to provide hypotheses for phase 2 and 3 clinical trials or cohort studies with regard to which potential biomarkers may warrant further investigation.
Point 2: Biopsies of metastatic sites are not described. Are they US or CT needle guided? Each biopsied metastatic site should be detailed in a table.
Response 2: Thank you for your valuable remark. We added a supplementary table 1 containing detailed information on the biopsies of metastatic sites and how they were taken.
Point 3: Data of OS and PFS should be deleted in that they are typical end points of phase 2 and 3 studies, again the analyzed series of patients is too small for this end points.
Response 3: We agree that this study is not adequately powered to analyse survival and we provided these values for exploratory purposes only. We have removed the figure and now only briefly mention the median OS and PFS with an explanation of its exploratory nature since readers might be interested in these numbers.
Point 4: In the “Materials and Methods Section” immunohistochemistry for the detection of “alpha smooth muscle actin (αSMA, DAKO M0851, 1:800 dilution) for staining of activated fibroblasts” is not detailed, there is no negative control in the experiments. In “Figure S2” magnification is lacking, the micrometric line bar is lacking. These are important bias of the experiments.
Response 4: Thank you for your valuable remark. The αSMA staining is a regularly performed staining in our hospital with proven specificity. Internal positive controls are fibroblasts. Other morphological structures do not stain and can be seen as internal negative controls. We added the magnification in Figure S2 (4x) and a micrometric line bar in each of the images.
Point 5: In the “Introduction Section” the Authors should specify that first line treatment usually consist of a doublet with a fluoropyrimidine and a platinum compound and with special regard to gastric cancer if the tumour is HER-2 positive trastuzumab should be added.
Response 5: We changed this sentence accordingly.
Point 6: At line 73 should be specified how many patients had received neoadjuvant chemoradiation with carboplatin, paclitaxel and fractionated radiation.
Response 6: This indeed is an important point. We added supplementary table 1 containing previous anti-cancer treatment for each patient.
Point 7: At line 74 the sentence “and six had undergone curative surgery” should be changed in “and six had undergone to potentially curative surgery”.
Response 7: The reviewer is certainly right and we changed this sentence accordingly.

Reviewer 2 Report
This manuscript is well described. I think there is no problem on the content.
Just one point, in line 83, the abbreviation "MTD" should be explained because of its first appearance.
Author Response
Response to Reviewer 2 Comments
Point 1: This manuscript is well described. I think there is no problem on the content. Just one point, in line 83, the abbreviation "MTD" should be explained because of its first appearance.
Response 1: Dear esteemed reviewer, thank you for your positive evaluation of our work. In the new version of the manuscript we added an explanation of the abbreviation MTD: maximum tolerated dose (line 37).
Reviewer 3 Report
To authors
The authors performed Phase I dose escalation study of the addition of nab-paclitaxel for the first line treatment of metastatic esophagogastric adenocarcinoma. Moreover, additional biomarker study was also planned. I think such a clinical trial is so important for developments of a new therapeutic strategy in order to improve the prognosis. I understand the feasible dose of this study. However, in the present study, there are some problems as following:
Major Comments
1. In the present study, many types of patients were enrolled. Therefore, clinicopathological features of the primary lesion and primary therapy, neo-adjuvant or adjuvant therapy, should be assessed in more detail. These data may affect to the results or discussion. Particularly, esophagectomy underwent in many patients compared to gastrectomy. Moreover, it should be shown which patients enrolled in which protocols.
2. In the present study, I cannot understand the importance of the biomarker study. Certainly, ADAM12 expression related with prognosis. However, significance or relationship of ADAM12 expression in this Phase I study are unclear. The differences of response to this protocol should be examined. Moreover, if the authors aimed to detect prognostic markers for this regimen, larger cohort study should be planned.
3. In the biomarker analysis, serum samples were obtained prior to initial dosing. However, these samples may include many different influences due to adjuvant therapies, such as operation, chemotherapy, or radiotherapy. The methods of examination or the candidates should be reconsidered.
Author Response
Response to Reviewer 3 Comments
Point 1: In the present study, many types of patients were enrolled. Therefore, clinicopathological features of the primary lesion and primary therapy, neo-adjuvant or adjuvant therapy, should be assessed in more detail. These data may affect to the results or discussion. Particularly, esophagectomy underwent in many patients compared to gastrectomy. Moreover, it should be shown which patients enrolled in which protocols.
Response 1: We thank the esteemed reviewer for this remark. As requested, we added supplementary table 1 containing clinicopathological features of the primary lesion and previous anti-cancer treatment for each patient. We also added a section in the discussion on the heterogeneity of our patient group.
“Furthermore, the heterogeneity of the patients studied in our trial is a limitation. This pertains not only to the primary site (esophagus, gastro-esophageal junction, and stomach), but also to previous anti-cancer treatment including prior surgery. Relatively more patients had a prior esophagectomy compared to a gastrectomy which may interfere with survival and susceptibility to toxicity, but to the best of our knowledge there is no definitive data on this difference. The relatively small number of patients in a phase I trial hampers further analyses of specific subgroups. However, in clinical practice, both pretreated and untreated patients and both esophageal and gastric cancer patients tend to receive the same first line palliative treatment in the Netherlands based on the results of the REAL-2 study, where these groups were studied collectively. One can argue the biomarker levels of pretreated patients could be lower than previously untreated patients because of lack of the primary tumor and it’s stroma, this has to be studied in larger cohorts.”
Regarding the discrepancy in prior esophagectomy compared to gastrectomy the reviewer raises a potentially relevant point. However, to the best of our knowledge there is no data on this specific difference.
Point 2: In the present study, I cannot understand the importance of the biomarker study. Certainly, ADAM12 expression related with prognosis. However, significance or relationship of ADAM12 expression in this Phase I study are unclear. The differences of response to this protocol should be examined. Moreover, if the authors aimed to detect prognostic markers for this regimen, larger cohort study should be planned.
Response 2: Thank you for your valuable remark. We agree that this study is not adequately powered to demonstrate true prognostic value of a biomarker. Nevertheless, we added these exploratory analyses because we believe it is important to investigate potential biomarkers in early phases of clinical trials to provide hypotheses for phase 2 and 3 clinical trials or cohort studies with regard to which potential biomarkers may warrant further investigation. Your suggestion to further analyse the response of the ADAM12 subgroups is a relevant one, yet we think these groups are too small in this phase 1 trial. No evident patterns in ADAM12 levels upon treatment with the ACTION regimen were seen in this patient group. We did compare baseline ADAM12 concentrations between response groups (CR, PR, SD and PD/non-evaluable) in supplementary figure 4. We fully agree with your suggestion to further study this biomarker in larger groups to assess it’s possible predictive value and therefore included the following sentence in the discussion.
“This prognostic, and perhaps even predictive value of ADAM12 needs to be further studied in larger cohorts with the possibility of multivariate testing - the lack of which is a clear limitation of our present study and underscores the exploratory nature of our findings.”
Point 3: In the biomarker analysis, serum samples were obtained prior to initial dosing. However, these samples may include many different influences due to adjuvant therapies, such as operation, chemotherapy, or radiotherapy. The methods of examination or the candidates should be reconsidered.
Response 3: The reviewer raises an important point. The following has been included in the results section of our manuscript.
“The median ADAM12 values between the pre-treated and previously untreated groups were statistically similar (pre-treated 171 pg/ml and previously untreated 348 pg/ml, Mann-Whitney p=0.90). Median ADAM12 values between patients who received prior surgery were also statistically similar to those of patients who had not received prior surgery (187 pg/ml vs 243 pg/ml, respectively, Mann-Whitney p=0.88).”
Round 2
Reviewer 1 Report
Accepted in the present form
Reviewer 3 Report
To Authors
The authors revised their manuscript according to the reviewers’ comments. However, I think the contents including the supplementary data will not be appropriate for Phase 1 study. Although I can understand the feasibility of additional dose of nab-paclitaxel, the registered cases or biomarker analysis should have been selected a little more. Therefore, I’m afraid that this manuscript will not be suitable for the publication to Cancers.
Major Comments
1. Again, I think the many types of patients’ background as shown in supplementary Table 1 are not appropriate for this study. The degree and sites of metastasis or differences of previous treatments will have major impact on the results.
2. The authors mentioned “the median ADAM12 values between the pre-treated and previously untreated groups were statistically similar”. However, even if there is no statistical difference, the median values are about 2 times different. I think this difference will be large in the study of many cases.